# Boost Neural Networks by Checkpoints

**Feng Wang[1], Guoyizhe Wei[1], Qiao Liu[2], Jinxiang Ou[1], Xian Wei[3], Hairong Lv[1,4] ***

[1]Department of Automation, Tsinghua University
[2]Department of Statistics, Stanford University
[3]Software Engineering Institute, East China Normal University
[4]Fuzhou Institute of Data Technology
{wangf19, wgyz19, ojx19}@mails.tsinghua.edu.cn
liuqiao@stanford.edu, xian.wei@tum.de
lvhairong@tsinghua.edu.cn

## Abstract

Training multiple deep neural networks (DNNs) and averaging their outputs is a simple way to improve the predictive performance. Nevertheless, the multiplied training cost prevents this ensemble method to be practical and efficient. Several recent works attempt to save and ensemble the checkpoints of DNNs, which only requires the same computational cost as training a single network. However, these methods suffer from either marginal accuracy improvements due to the low diversity of checkpoints or high risk of divergence due to the cyclical learning rates they adopted. In this paper, we propose a novel method to ensemble the checkpoints, where a boosting scheme is utilized to accelerate model convergence and maximize the checkpoint diversity. We theoretically prove that it converges by reducing exponential loss. The empirical evaluation also indicates our proposed ensemble outperforms single model and existing ensembles in terms of accuracy and efficiency. With the same training budget, our method achieves 4.16% lower error on Cifar-100 and 6.96% on Tiny-ImageNet with ResNet-110 architecture. Moreover, the adaptive sample weights in our method make it an effective solution to address the imbalanced class distribution. In the experiments, it yields up to 5.02% higher accuracy over single EfficientNet-B0 on the imbalanced datasets.

## 1 Introduction

DNN ensembles often outperform individual networks in either accuracy or robustness [Hansen and Salamon, 1990, Zhou et al., 2002]. Particularly, in the cases when memory is restricted, or complex models are difficult to deploy, ensembling light-weight networks is a good alternative to achieve the performance comparable with deep models. However, since training DNNs is computationally expensive, straightforwardly ensembling multiple networks is not acceptable in many real-world scenarios.

A trick to avoid the exorbitant computational cost is to fully utilize the middle stages — or so-called checkpoints — in one training process, instead of training additional networks. Here we refer to this technique as Checkpoint Ensemble (CPE). Despite its merit of no additional training computation, an obvious problem of CPE is that the checkpoints sampled from one training process are often very similar, which violates the consensus that we desire the base models in an ensemble are accurate but sufficiently diverse. To enhance the diversity, conventional ensembles often differ the base models from initialization, objective function, or hyperparameters [Breiman, 1996, Freund and Schapire, 1997, Zhou et al., 2002], whilst the recent CPE methods try to achieve this goal by cyclical learning

---

*Corresponding author

rate scheduling, with the assumption that the high learning rates are able to force the model jumping out of the current local optima and visiting others [Huang et al., 2017a, Garipov et al., 2018, Zhang et al., 2020]. However, this effect is not theoretically promised and the cyclically high learning rates may incur oscillation or even divergence when training budget is limited.

In this paper, we are motivated to efficiently exploit the middle stages of neural network training process, obtaining significant performance improvement only at the cost of additional storage, meanwhile the convergence is promised. We design a novel boosting strategy to ensemble the checkpoints, named CBNN (**C**heckpoint-**B**oosted **N**eural **N**etwork). In contrast to cyclical learning rate scheduling, CBNN encourages checkpoints diversity by adaptively reweighting training samples. We analyze its advantages in Section 4 from two aspects. Firstly, CBNN is theoretically proved to be able to reduce the exponential loss, which aims to accelerate model convergence even if the network does not have sufficiently low error rate. Secondly, reweighting training samples is equivalent to tuning the loss function. Therefore, each reweighting process affects the distribution of local minimum on the loss surface, and thereby changes the model's optimization direction, which explains the high checkpoint diversity of CBNN.

Moreover, the adaptability of CBNN makes it effective in tackling with imbalanced datasets (datasets with imbalanced class distribution). This is due to CBNN's ability to automatically allocate high weights to the "minority-class" samples, while most commonly used imbalanced learning methods require manually assigning sample weights [Buda et al., 2018, Johnson and Khoshgoftaar, 2019].

## 2 Related Work

In general, training a collection of neural networks with conventional ensemble strategies such as Bagging [Breiman, 1996] and Boosting [Freund and Schapire, 1997] improves the predictors on accuracy and generalization. For example, [Moghimi et al., 2016] proposed the BoostCNN algorithm, where a finite number of convolutional neural networks (CNNs) are trained based on a boosting strategy, achieving a superior performance than a single model. Several other strategies such as assigning different samples to the base models [Zhou et al., 2018] or forming ensembles with shared low-level layers of neural networks [Zhu et al., 2018] also combine the merits of DNNs and ensemble learning.

However, these methods are computationally expensive. Alternatively, the "implicit ensemble" techniques [Wan et al., 2013, Srivastava et al., 2014, Huang et al., 2016] or adding random noise layers [Liu et al., 2018] avoid this problem. Such methods are sometimes seen as regularization approaches and can work in coordination with our proposed method. Recently, checkpoint ensemble has become increasingly popular as it improves the predictors "for free" [Huang et al., 2017a, Zhang et al., 2020]. It was termed as "Horizontal Voting" in [Xie et al., 2013], where the outputs of the checkpoints are straightforwardly ensembled as the final prediction. However, these checkpoints share a big intersection of misclassified samples which only bring very marginal improvements.

Therefore, the necessary adjustments to the training process should be made to enhance this diversity. For instance, [Huang et al., 2017a] proposed the Snapshot Ensemble method, adopting the cyclic cosine annealing learning rate to enforce the model jumping out of the current local optima. Similarly, another method FGE (Fast Geometric Ensembling) [Garipov et al., 2018] copies a trained model and further fine-tunes it with a cyclical learning rate, saving checkpoints and ensembling them with the trained model. More recently, [Zhang et al., 2020] proposed the Snapshot Boosting, where they modified the learning rate restarting rules and set different sample weights during each training stage to further enhance the diversity of checkpoints. Although there are weights of training samples in Snapshot Boosting, these weights are updated only after the learning rate is restarted and each update begins from the initialization. Such updates are not able to significantly affect the loss function, hence it is essentially still a supplementary trick for learning rate scheduling to enhance the diversity.

In this paper, our proposed method aims to tune the loss surface during training by the adaptive update of sample weights, which induces the network to visit multiple different local optima. Our experimental results further demonstrate that our method, CBNN significantly promotes the diversity of checkpoints and thereby achieves the highest test accuracy on a number of benchmark datasets with different state-of-the-art DNNs.

---

**Algorithm 1** Checkpoint-Boosted Neural Networks

---

**Input**: Number of training samples: $n$; number of classes: $k$; deviation rate: $\eta$;
total training iterations: $T$; training iterations per checkpoint: $t$
**Require**: Estimated weight of the trained model: $\lambda_0$
**Output**: An ensemble of networks $\{G_1(x), G_2(x) \dots\}$ (Note that $G_j(x) \in \mathbb{R}^k$, $G_j^y(x) = 1$ if $G_j(x)$
predicts $x$ belonging to the $y$-th class, otherwise $G_j^y(x) = 0$)

1: Initialize $m \leftarrow 1$      // number of base models
2: Uniformly initialize the sample weights:
$$\omega_{mi} \leftarrow 1/n,\ i \leftarrow 1, ..., n$$
3: **while** $T - mt > 0$ & $\sum_{j=0}^{m-1} \lambda_j < 1/\eta$ **do**
4:      Train model for $\min(t, T - mt)$ iterations, get $G_m(x)$
5:      Calculate the weighted error rate on training set:
$$e_m \leftarrow \sum_{i=1}^{n} \omega_{mi} I(G_m^{y_i}(x_i) = 0)$$
6:      Calculate the weight of $m$-th checkpoint:
$$\lambda_m \leftarrow \log((1 - e_m)/e_m) + \log(k - 1)$$
7:      Update the sample weights:
$$\omega_{m+1,i} \leftarrow \omega_{mi} \exp\left(-\eta \lambda_m G_m^{y_i}(x_i)\right) / \sum_{i=1}^{n} \omega_{mi} \exp\left(-\eta \lambda_m G_m^{y_i}(x_i)\right)$$
8:      Save the checkpoint model at current step.
9:      $m \leftarrow m + 1$
10: **end while**
11: Train model for $\max(t, T - (m - 1)t)$ iterations, get $G_m(x)$.
12: Execute step 5 and 6 to calculate $\lambda_m$.
13: **return** $G(x) = \sum_{j=1}^{m} \lambda_j G_j(x) / \sum_{j=1}^{m} \lambda_j$

---

## 3 Methodology

### 3.1 Overview of Checkpoint-Boosted Neural Networks

CBNN aims to accelerate convergence and improve accuracy by saving and ensembling the middle stage models during neural networks training. Its ensemble scheme is inspired by boosting strategies [Freund and Schapire, 1997, Hastie et al., 2009, Mukherjee and Schapire, 2013, Saberian and Vasconcelos, 2011], whose main idea is to sequentially train a set of base models with weights for training samples. After one base model is trained, it enlarges the weights of the misclassified samples (meanwhile, the weights of the correctly classified samples are reduced after re-normalization). Intuitively, this makes the next base model focus on the samples that were previously misclassified. In testing phase, the output of boosting is the weighted average of the base models' predictions, where the weights of the models depend on their error rates on the training set.

We put the sample weights on the loss function, which is also reported in many cost-sensitive learning and imbalanced learning methods [Buda et al., 2018, Johnson and Khoshgoftaar, 2019]. In general, given a DNN model $p \in \mathbb{R}^{|G|}$ and training dataset $\mathcal{D} = \{(x_i, y_i)\}(|\mathcal{D}| = n, x_i \in \mathbb{R}^d, y_i \in \mathbb{R})$ where $|G|$ is the number of model parameters and $x_i, y_i$ denote the features and label of the $i$-th sample, the weighted loss is formulated as

$$\mathcal{L} = \frac{\sum_{i=1}^{n} \omega_i l(y_i, f(x_i; p))}{\sum_{i=1}^{n} \omega_i} + r(p), \tag{1}$$

where $l$ is a loss function (e.g. cross-entropy), $f(x_i; p)$ is the output of $x_i$ with the model parameters $p$, and $r(p)$ is the regularization term.

The next section introduces the training procedure of CBNN, including the weight update rules and ensemble approaches, which are also summarized in Algorithm 1.

### 3.2 Training Procedure

We start training a network with initializing the sample weight vector as a uniform distribution:

$$\Omega_1 = [\omega_{1,1}, ..., \omega_{1,i}, ..., \omega_{1,n}],\ \omega_{1,i} = \frac{1}{n}, \tag{2}$$

where $n$ denotes the number of training samples. After every $t$ iterations of training, we update the weights of training samples and save a checkpoint, where the weight of the $m$-th checkpoint is calculated by

$$\lambda_m = \log\frac{1 - e_m}{e_m} + \log(k - 1), \tag{3}$$

in which $k$ is the number of classes, and

$$e_m = \sum_{i=1}^{n} \omega_{mi} I(G_m^{y_i}(x_i) = 0) \tag{4}$$

denotes the weighted error rate. Note that $G_m(x) \in \mathbb{R}^k$ yields a one-hot prediction, i.e., given the predicted probabilities of $x$, $[p^y(x)]_{y=1}^{k}$,

$$G_m^y(x) = \begin{cases} 1, & p^y(x) > p^{y'}(x), \ \forall y' \neq y, y' \in [1, k] \\ 0, & \text{else} \end{cases}. \tag{5}$$

Briefly, $G_m^y(x) = 1$ implies that the model $G_m(\cdot)$ correctly predict the sample $x$ and $G_m^y(x) = 0$ implies the misclassification. According to Equation 3, the weight of a checkpoint increases with the decrease of its error rate. The term $\log(k-1)$ may be confusing, which is also reported in the multi-class boosting algorithm SAMME [Hastie et al., 2009]. Intuitively, $\log(k-1)$ makes the weight of the base models $\lambda_m$ positive, since the error rate in a $k$-classification task is no less than $(k-1)/k$ (random choice is able to obtain $1/k$ accuracy). In fact, the positive $\lambda_m$ is essential for convergence, which will be further discussed in Section 4.1.

We then update the sample weights with the rule that the weight of the sample $(x_i, y_i)$ increases if $G_m^{y_i}(x_i) = 0$ (i.e., it is misclassified) and decrease if $G_m^{y_i}(x_i) = 1$, which is in accordance with what we introduced in Section 3.1. They are updated by

$$\omega_{m+1,i} = \frac{\omega_{mi}}{Z_m} \exp\left(-\eta\lambda_m G_m^{y_i}(x_i)\right), \tag{6}$$

where $\eta$ is a positive hyperparameter named "deviation rate" and

$$Z_m = \sum_{i=1}^{n} \omega_{mi} \exp\left(-\eta\lambda_m G_m^{y_i}(x_i)\right) \tag{7}$$

is a normalization term to keep $\sum_{i=1}^{n} \omega_{mi} = 1$. It is very important to consider the tradeoff of the deviation rate $\eta$. Here we firstly present an intuitive explanation. In Equation 6, a high deviation rate speeds up the weight update, which seems to help enhance the checkpoint diversity. However, this will lead to very large weights until some specific samples dominate the training set. From grid search, we find that setting $\eta = 0.01$ achieves significant performance improvement.

In Section 4.1, we are going to prove that CBNN reduces the exponential loss with the condition

$$\sum_{m=1}^{M} \lambda_m < 1/\eta, \tag{8}$$

i.e., the sum of the weights of $M - 1$ checkpoints $\lambda_m$ ($m = 1, 2, \ldots, M - 1$) and the finally trained model $\lambda_M$ should be limited. To meet this condition, therefore, sometimes we need to limit the number of checkpoints. However, we cannot acquire $\lambda_M$ beforehand. A simple way to address this issue is to estimate the error rate of the finally trained model and calculate its weight (denoted by $\lambda_0$) according to Equation 3. For example, if regard that the error rate of the network in a 100-class task is not less than 0.05, we can estimate $\lambda_0$ as $\log(0.95/0.05) + \log(99) = 7.54$. Then, we stop saving checkpoints and updating the sample weights until $\sum_{j=0}^{m} \lambda_j > 1/\eta$.

The final classifier is an ensemble of the finally trained model $G_M(X)$ and $M - 1$ checkpoints $G_m(X)$, $m = 1, 2, \ldots, M - 1$:

$$G(x) = \sum_{m=1}^{M} \lambda_m G_m(x) / \sum_{m=1}^{M} \lambda_m. \tag{9}$$

For saving storage and test time, we can select part of the $M - 1$ checkpoints to ensemble, while selecting with equal interval is expected to achieve high diversity.

# 4 Theoretical Analyses

## 4.1 Convergence of CBNN

According to Equation 9, the output of the final model, $G(x)$, is the weighted average output of the base model set $\{G_m(x)\}_{m=1}^M$, where $G^y(x) \in [0,1]$ denotes the predicted probability that the sample $x$ belongs to the $y$-th category. Therefore, we desire as high $G^{y_i}(x_i)$ as possible on each sample $x_i$ and the exponential loss

$$\mathcal{L}_{exp} = \frac{1}{n} \sum_{i=1}^{n} \exp(-G^{y_i}(x_i)) \tag{10}$$

is often used to measure the error of the model. We analyze the convergence of our method by presenting the upper bound of this exponential loss in CBNN and proving that this bound decreases as the number of base models, $M$, increases.

**Theorem 1.** *The exponential loss $\mathcal{L}_{exp}$ in CBNN is bounded above by $\mathcal{L}_{exp} \leqslant \prod_{m=1}^M Z_m$.*

*Proof.*

$$
\begin{aligned}
\mathcal{L}_{exp} &= \frac{1}{n} \sum_{i=1}^{n} \exp\left(-\sum_{m=1}^{M} \lambda_m G_m^{y_i}(x_i) \Big/ \sum_{m=1}^{M} \lambda_m\right) \leqslant \frac{1}{n} \sum_{i=1}^{n} \exp\left(-\eta \sum_{m=1}^{M} \lambda_m G_m^{y_i}(x_i)\right) \\
&= \frac{1}{n} \sum_{i=1}^{n} \prod_{m=1}^{M} \exp\left(-\eta \lambda_m G_m^{y_i}(x_i)\right) = \sum_{i=1}^{n} \omega_{1,i} \exp\left(-\eta \lambda_1 G_1^{y_i}(x)\right) \prod_{m=2}^{M} \exp\left(-\eta \lambda_m G_m^{y_i}(x_i)\right) \\
&= Z_1 \sum_{i=1}^{n} \omega_{2,i} \exp\left(-\eta \lambda_2 G_2^{y_i}(x)\right) \prod_{m=3}^{M} \exp\left(-\eta \lambda_m G_m^{y_i}(x_i)\right) \\
&= Z_1 \dots Z_{M-1} \sum_{i=1}^{n} \omega_{M,i} \exp\left(-\eta \lambda_M G_M^{y_i}(x)\right) = \prod_{m=1}^{M} Z_m
\end{aligned}
\tag{11}
$$

**Theorem 2.** *The upper bound of $\mathcal{L}_{exp}$ in CBNN decreases as $M$ increases.*

*Proof.*

$$
\begin{aligned}
Z_m &= \sum_{i=1}^{n} \omega_{mi} \exp\left(-\eta \lambda_m G_m^{y_i}(x_i)\right) = \sum_{G_m^{y_i}(x_i)=1} \omega_{mi} \exp\left(-\eta \lambda_m\right) + \sum_{G_m^{y_i}(x_i)=0} \omega_{mi} \\
&= (1 - e_m) \exp\left(-\eta \lambda_m\right) + e_m < 1
\end{aligned}
\tag{12}
$$

Note that, according to Formula 12, $Z_m < 1$ when $\lambda_m$ is positive, which explains why we add $\log(k-1)$ in Equation 3. **Theorem 1** and **2** demonstrate an exponential decrease on $\mathcal{L}_{exp}$ bound as we save more base models. Moreover, according to Formula 12, even a moderately low error rate $e_m$ of the base model can lead to a significant decrease on the $\mathcal{L}_{exp}$ upper bound.

## 4.2 Diversity of Checkpoints

We also demonstrate the superior performance of CBNN from the viewpoint of checkpoint diversity. As reported in the prior works [Auer et al., 1996, Dauphin et al., 2014], there is a large number of local optima and saddle points on a DNN loss surface. However, even the nearby optima may yield a small intersection of misclassified samples from a given dataset [Garipov et al., 2018].

When we enlarge the weights of the misclassified samples corresponding to a local optimum, the loss $\mathcal{L}$ of this optimum increased, whereas it may be only slightly changed (or even decreased) around the other optima. Based on this observation, CBNN adopts the loss function in the form of Equation 1 and tunes the loss surface adaptively by increasing the weights of the misclassified samples during training. This process enlarges the loss of the current local optimal domain, making it easier for the model to jump out of the current optimum and visit more other optima. In other

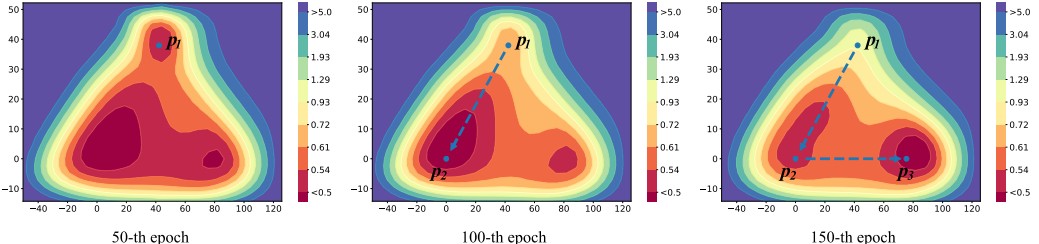

Figure 1: The moving $L_2$-regularized cross-entropy of an EfficientNet-B0 on Cifar-100 in CBNN, where $p_1$, $p_2$, and $p_3$ are three local optima of the model. The network tends to converge to $p_1$ at the beginning (the 50-th epoch, left panel). As the loss surface moves, it converges to $p_2$ (the 100-th epoch, middle panel) and finally to $p_3$ (the 150-th epoch, right panel). As it is seen the moving loss surface makes checkpoints accurate but diverse, hence enhances their value to be ensembled.

words, the checkpoints of this model have higher diversity and ensembling them is expected to yield considerable performance improvements.

Fig.1 visualizes the $L_2$-regularized cross-entropy loss on Cifar-100 [Krizhevsky et al., 2009] of an EfficientNet-B0 [Tan and Le, 2019] trained with CBNN strategy, where the three panels respectively display the loss surface at the 50, 100, 150-th epoch in a two-dimensional subspace view [1]. It is shown that the loss surface has significantly changed during the training process, which thereby induces the network to converge to different local optima at different training stages. Specifically, the model tends to converge to one local optimum at the 50-th epoch ($p_1$, Fig.1 Left), whereas the loss around $p_1$ is gradually increased and the model is then forced to converge to $p_2$ at the 100-th epoch (Fig.1 Middle). As the loss surface is further modified, the model leaves $p_2$ and finally converges to $p_3$ (Fig.1 Right).

## 5   Experiments

We compare the effectiveness of CBNN with competitive baselines in this section. All the experiments are conducted on four benchmark datasets: **Cifar-10**, **Cifar-100** [Krizhevsky et al., 2009], **Tiny ImageNet** [Le and Yang, 2015], and **ImageNet** ILSVRC 2012 [Deng et al., 2009].

### 5.1   Experiment Setup

**DNN Architectures.** We test the ensemble methods on the three commonly used network architectures, including ResNet [He et al., 2016], DenseNet [Huang et al., 2017b] and EfficientNet [Tan and Le, 2019]. We adopt an 110-layer ResNet and 40-layer DenseNet introduced in [He et al., 2016] and [Huang et al., 2017b], with $32 \times 32$ input size. For EfficientNet, we use EfficientNet-B0 and B3 with images resized to $224 \times 224$ pixels. Despite EfficientNet-B3 suggested a larger image size [Tan and Le, 2019], $224 \times 224$ would be sufficient to achieve high performance while reducing memory footprint and computational cost. All the models are trained with 0.2 dropout rate (0.3 for EfficientNet-B3) and images are augmented by AutoAugment [Cubuk et al., 2019].

**Baseline Methods.** Firstly, we train a **Single Model (referred to as Single)** as baseline, adopting a standard decaying learning rate that is initialized to 0.05 and drops by 96% every two epochs with five epochs warmup [Gotmare et al., 2018]. Here we compare CBNN with state-of-the-art checkpoint ensemble models, including **Snapshot Ensemble (SSE)** [Huang et al., 2017a], **Snapshot Boosting (SSB)** [Zhang et al., 2020], and **Fast Geometric Ensemble (FGE)** [Garipov et al., 2018]. In Snapshot Ensemble, the learning rate scheduling rules follow [Huang et al., 2017a] and we set $\alpha = 0.2$, which achieves better performance in our experiments. Similarly, we set $r_1 = 0.1$, $r_2 = 0.5$,

---

[1] $p_1, p_2, p_3 \in \mathbb{R}^{|G|}$ ($|G|$ is the number of model parameters) are parameter vectors of the model iterated to 50, 100 and 150 epochs, respectively. We adopt a similar affine transformation as [Garipov et al., 2018], setting $u = p_3 - p_2$, $v = (p_1 - p_2) - <p_1 - p_2, p_3 - p_2> / ||p_3 - p_2||^2 \cdot (p_3 - p_2)$, and a point $(x, y)$ in each panel denotes the vector $p = p_2 + x \cdot u/||u|| + y \cdot v/||v||$.

Table 1: Error rates (%) on the four datasets with different ensemble methods. We repeat these experiments five times to get the average test error and standard deviation. Note that we omit the deviations of Cifar-10 since they are relatively low and similar. The best results of each model and dataset are **bolded**.

| Model | Method | Cifar-10 | Cifar-100 | Tiny-ImageNet | ImageNet |
|---|---|---|---|---|---|
| | Single | 5.63 | $27.67 \pm 0.25$ | $45.10 \pm 0.36$ | - |
| | SSE | 5.41 | $24.27 \pm 0.26$ | $39.08 \pm 0.34$ | - |
| ResNet-110 | SSB | 5.61 | $27.09 \pm 0.35$ | $43.68 \pm 0.49$ | - |
| | FGE | 5.50 | $26.21 \pm 0.23$ | $41.16 \pm 0.31$ | - |
| | CBNN | **5.25** | $\mathbf{23.51 \pm 0.20}$ | $\mathbf{38.14 \pm 0.22}$ | - |
| | Single | 5.13 | $23.15 \pm 0.22$ | $38.24 \pm 0.34$ | - |
| | SSE | **4.78** | $21.45 \pm 0.29$ | $36.35 \pm 0.30$ | - |
| DenseNet-40 | SSB | 5.13 | $22.59 \pm 0.31$ | $37.99 \pm 0.44$ | - |
| | FGE | 4.82 | $22.07 \pm 0.28$ | $37.72 \pm 0.27$ | - |
| | CBNN | **4.78** | $\mathbf{20.66 \pm 0.17}$ | $\mathbf{34.28 \pm 0.20}$ | - |
| | Single | 3.88 | $20.87 \pm 0.25$ | $34.50 \pm 0.35$ | $23.70 \pm 0.21$ |
| | SSE | 3.56 | $19.70 \pm 0.25$ | $33.02 \pm 0.32$ | $23.51 \pm 0.24$ |
| EfficientNet-B0 | SSB | 3.87 | $20.12 \pm 0.41$ | $34.02 \pm 0.46$ | $23.75 \pm 0.35$ |
| | FGE | 3.68 | $19.58 \pm 0.24$ | $32.76 \pm 0.31$ | $23.43 \pm 0.22$ |
| | CBNN | **3.37** | $\mathbf{18.06 \pm 0.20}$ | $\mathbf{31.16 \pm 0.26}$ | $\mathbf{22.69 \pm 0.19}$ |
| | Single | 3.45 | $17.28 \pm 0.22$ | $31.86 \pm 0.30$ | $18.30 \pm 0.15$ |
| | SSE | 3.30 | $16.84 \pm 0.29$ | $31.01 \pm 0.31$ | $19.16 \pm 0.22$ |
| EfficientNet-B3 | SSB | 3.43 | $17.25 \pm 0.32$ | $31.49 \pm 0.42$ | $19.59 \pm 0.34$ |
| | FGE | **3.27** | $16.96 \pm 0.24$ | $30.97 \pm 0.30$ | $18.68 \pm 0.20$ |
| | CBNN | 3.28 | $\mathbf{16.07 \pm 0.13}$ | $\mathbf{29.85 \pm 0.23}$ | $\mathbf{17.55 \pm 0.14}$ |

$p_1 = 2$ and $p_2 = 6$ for Snapshot Boosting. In Fast Geometric Ensemble (FGE), we train the first model with standard learning rate (the same as Single Model) and the other models with the settings according to [Garipov et al., 2018], and set $\alpha_1 = 5 \cdot 10^{-2}$, $\alpha_2 = 5 \cdot 10^{-4}$ to all the datasets and DNN architectures. Our method, **CBNN** adopts the learning rate used in training Single Model as well, and setting $\eta = 0.01$.

**Training Budget.** Checkpoint ensembles are designed to save training costs, therefore, to fully evaluate their effects, the networks should be trained for relatively fewer epochs. In our experiments, we train the DNNs from scratch for 200 epochs on Cifar-10, Cifar-100, Tiny-ImageNet and 300 epochs on ImageNet. We save six checkpoint models for SSE and FGE, however, considering the large scale and complexity of ImageNet, saving three base models is optimal for SSE and FGE since more learning rate cycles may incur divergence. This number of SSB is automatically set by itself. Although our method supports a greater number of base models which leads to higher accuracy, for clear comparison, we only select the same number of checkpoints as SSE and FGE.

## 5.2 Evaluation of CBNN

The evaluation results are summarized in Table 1, where all the methods listed are at the same training cost. It is seen that in most cases, CBNN achieves the lowest error rate on the four tested datasets with different DNN architectures. Our method achieves a significant error reduction on Cifar-100 and Tiny-ImageNet. For instance, with EfficientNet-B0 as the base model, using CBNN decreases the mean error rate by 2.81% and 3.34%, on Cifar-100 and Tiny-ImageNet, respectively. In the experiments, networks tend to converge with a lower speed on ImageNet mainly due to the higher level of complexity and a larger number of training images in this dataset. Although we reduce the learning rate cycles of SSE and FGE (SSB automatically restarts the learning rate), these ensemble methods only slightly increase the accuracy on ImageNet with EfficientNet-B3.

We quantitatively evaluate the efficiency of ensemble methods by comparing the GPU hours they take to achieve a certain test accuracy for a given DNN and dataset. For the checkpoint ensemble methods, the test accuracy is calculated by the ensemble of current model and saved checkpoints. For

Table 2: GPU hours required for EfficientNet-B0 to reach certain test accuracy. We repeat these experiments five times to get average GPU hours and standard deviation. Note that we omit the deviations of Cifar-10/60% since they are relatively low and similar. The best results are **bolded**.

| Method | Cifar-10 | | | Cifar-100 | | |
|--------|------|------|------|------|------|------|
| | 60% | 80% | 90% | 40% | 60% | 70% |
| Single | 0.38 | $0.78 \pm 0.12$ | $5.05 \pm 0.55$ | $0.91 \pm 0.08$ | $2.71 \pm 0.12$ | $7.47 \pm 0.79$ |
| SSE | 0.38 | $0.75 \pm 0.10$ | $3.92 \pm 0.21$ | $0.87 \pm 0.08$ | $2.54 \pm 0.11$ | $5.16 \pm 0.31$ |
| SSB | 0.39 | $0.76 \pm 0.10$ | $4.22 \pm 0.38$ | $0.99 \pm 0.09$ | $2.62 \pm 0.12$ | $5.67 \pm 0.30$ |
| PE | 0.72 | $2.13 \pm 0.34$ | $10.13 \pm 0.78$ | $2.11 \pm 0.11$ | $7.78 \pm 1.23$ | $17.32 \pm 1.30$ |
| CBNN | **0.36** | $\mathbf{0.68 \pm 0.07}$ | $\mathbf{3.62 \pm 0.15}$ | $\mathbf{0.85 \pm 0.07}$ | $\mathbf{2.33 \pm 0.10}$ | $\mathbf{4.82 \pm 0.22}$ |

example, if we save a checkpoint every 5,000 iterations, the test accuracy of the 12,000-th iteration is calculated by the averaged outputs of the 5,000, 10,000 and 12,000-th iteration. Additionally, we train four models in parallel with different random initialization and add up their training time to evaluate the efficiency of conventional ensembles (**referred to as Parallel Ensemble, PE**). Table 2 summarizes the time consumption of different methods on Nvidia Tesla P40 GPUs, where the checkpoint ensembles are shown to be more efficient. In particular, CBNN reaches 0.7 test accuracy on Cifar-100 costing only 64.5% training time of the single model. Although the conventional ensembles are able to improve accuracy, in our experiments, the parallel ensemble method often results in 2-3 times training overhead than single model to achieve the same accuracy.

## 5.3 Diversity of Base Models

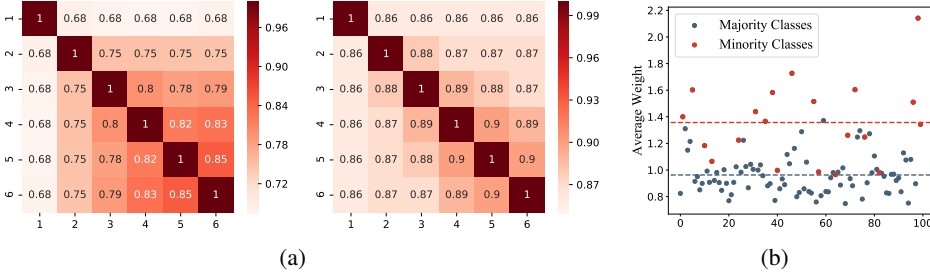

(a)                                                                                                  (b)

Figure 2: Experimental results of diversity and imbalanced datasets. **(a):** Pairwise correlation between the softmax outputs of any two checkpoint models of EfficientNet-B0 on Cifar-100 testing set. *Left:* CBNN. *Right:* Snapshot Ensemble. **(b):** Average weights over classes on imbalanced Cifar-100 (randomly select 20 classes and remove 450 training images from each of them). The total 100 points represent the 100 classes in Cifar-100 and the average weight of training samples in each class is measured by $y$-axis. Note that the average value of the red points (red dash line) is significantly higher than that in dark blue (dark blue dash line).

To measure the diversity of the base models, we compute the pairwise correlation between the softmax outputs of any two checkpoints and then compare our method with Snapshot Ensemble. The results on Cifar-100 with EfficientNet-B0 are shown in Fig.2(a), where the correlation values of CBNN (the left panel) are significantly lower than that of Snapshot Ensemble (the right panel). These lower correlation values of CBNN indicate that any two checkpoints predict with a smaller intersection of misclassified samples, which in other words implies a higher diversity.

The high diversity, as well as the visualization of the moving loss surface in Fig.1 are explained based on the following two aspects: (1) As it is introduced in Section 4.2, the weight update processes in our method change the shape of the loss surface, thus inducing the model to visit more nearby local optima; (2) We optimize the model in a mini-batch manner, where it takes a large step size of gradient decent in training the batch that contains high weighted samples, which increases the probability of the model to jump out of local optima.

## 5.4 Effect on Imbalanced Datasets

Table 3: Error rates (%) on imbalanced Cifar-100 and Tiny-ImageNet (randomly selected 20% classes and removed 90% training images from each of them). tsh and oversample denote Thresholding and Oversampling introduced in section 5.4. We repeat these experiments five times to get average values and present standard deviations of oversampling since this method introduces higher randomness. The best results of each model and dataset are **bolded**.

| Model | Method | Cifar-100 | | | Tiny-ImageNet | | |
|---|---|---|---|---|---|---|---|
| | | original | tsh | oversample | original | tsh | oversample |
| Efficient Net−B0 | Single | 29.19 | 26.93 | 27.55 ± 0.30 | 45.25 | 42.58 | 43.12 ± 0.67 |
| | SSE | 28.27 | 25.88 | 26.14 ± 0.42 | 43.79 | 41.26 | 42.06 ± 0.65 |
| | SSB | 28.66 | 26.51 | 27.18 ± 0.33 | 44.98 | 42.47 | 43.13 ± 0.69 |
| | FGE | 28.39 | 26.43 | 26.83 ± 0.29 | 43.77 | 41.03 | 41.78 ± 0.55 |
| | CBNN | **24.39** | - | - | **40.23** | - | - |
| Efficient Net−B3 | Single | 27.08 | 25.01 | 25.86 ± 0.30 | 43.34 | 41.33 | 42.01 ± 0.60 |
| | SSE | 26.49 | 24.56 | 25.11 ± 0.33 | 42.26 | 40.47 | 40.36 ± 0.45 |
| | SSB | 26.78 | 24.89 | 25.46 ± 0.39 | 42.99 | 41.30 | 41.76 ± 0.59 |
| | FGE | 26.23 | 24.16 | 24.86 ± 0.34 | 42.58 | 41.29 | 41.12 ± 0.48 |
| | CBNN | **22.98** | - | - | **39.02** | - | - |

The imbalanced class distribution often negatively impacts the performance of classification models. There are many existing methods tackling with this issue. As [Buda et al., 2018] claimed, most of these methods are mathematically equivalent. Thus, we implement two effective and commonly used approaches introduced in [Buda et al., 2018], **Random Minority Oversampling** and **Thresholding with Prior Class Probabilities**, as baselines.

Further, we construct "step imbalance" datasets based on Cifar-100 and Tiny-ImageNet in the way introduced in [Buda et al., 2018], i.e., we randomly select $k_{min}$ of the total $k$ classes as minority classes, setting $\mu = k_{min}/k$, $\rho = n_{max}/n_{min}$ where $n_{max}$, $n_{min}$ denote the number of training samples in majority and minority classes. We then randomly subsample the selected minority classes to match $\rho$. Here we set $\mu = 0.2$, $\rho = 10$.

Error rates on the imbalanced datasets are summarized in Table 3, where it is seen that the single model and the prior checkpoint ensembles suffer from significant error increase relative to training with the entire dataset (See Table 1), even if Oversampling or Thresholding are adopted to help rebalancing the dataset. In contrast, the imbalance processing only leads to a relatively low reduction on CBNN accuracy.

To further study how the sample weights are adapted to the imbalanced datasets, we obtain the average weights in each class after the CBNN model is trained. As it is seen in Fig.2(b), the average weights of the minority classes (the red points) are almost at a higher level than the majority (the dark blue points), which indicates that CBNN can adaptively set higher weights for the minority classes to balance the dataset.

## 6 Conclusion

Proper ensemble strategies improve DNNs in both accuracy and robustness. However, the high training cost of DNNs urges the ensembles to be sufficiently efficient for practical use. Checkpoint Ensembles often outperform the single model by exploiting the value of middle stage models during training, which do not require additional training cost. Our new proposed checkpoint ensemble method, CBNN, is theoretically proven to reduce the exponential loss and shown to yield more diverse base models, compared to the existing methods. A series of experiments demonstrate the superior performance of CBNN in both accuracy and efficiency. In particular, when training DNNs on a large scale dataset with limited computation budget, the prior checkpoint ensemble methods based on cyclical learning rate scheduling result in poor performance (e.g. training EfficientNet-B0 on ImageNet in Table 1). Moreover, on imbalanced datasets, CBNN is able to automatically rebalance

the class distribution by adaptive sample weights, which outperforms the commonly used methods with manually rebalancing approaches.

## Acknowledgement

This work was supported by the National Nature Science Foundation of China under Grant No. 42050101 and U1736210.

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
