# OpenReview forum: "Boost Neural Networks by Checkpoints"
_NeurIPS.cc/2021/Conference — NeurIPS 2021 Poster_

### Official Review · Reviewer_Bmco · 2021-07-17

**Rating:** 7
**Confidence:** 3

**Summary:**

This paper proposed a novel method to obtain a diverse set of checkpoint models for efficient ensembles. Through the boosting inspired sampling weighting scheme, the proposed method not only facilitates the convergence of the model but also produces checkpoints with diverse predictions with an enhanced ensemble performance. The authors of the paper theoretically prove the convergence of the proposed method and empirically demonstrate that the method is capable of producing ensembles with good performance given a fixed computational budget.

**Limitations And Societal Impact:**

 The authors adequately addressed the limitations and potential negative societal impact of their work.

**Main Review:**

All in all, this paper is well written and easy to follow. The proposed method seems technically sound. While the idea of sample reweighting is definitely not new in general in the context of deep learning (for example, see [1][2]), I personally find its connection to boosting and also the use of it as a way to promote model diversity among different checkpoints to be very interesting. Specifically, despite the reduction in training time computational cost, most of the previously proposed methods for "checkpoints ensembling" still perform poorly when compared to standard ensemble strategies likely due to low diversity among the models. The authors of the paper directly tackle this problem by utilizing a sample reweighting approach to introduce more diversity to the checkpoints. The authors of the paper also conduct a good amount of experiments and ablation to demonstrate the effectiveness of the proposed method.

Here are some shortcomings/suggestions of the paper. Firstly, in my opinion, as an illustrative purpose, It might be worthwhile to see how big the gap is in terms of accuracy and diversity when compared to an explicit ensemble. In addition, it might be interesting to also give a more careful analysis (in addition to the ones present in section 5.3) on why the proposed method outperforms previously proposed checkpoint ensemble methods. Is it because of larger diversity or better individual model performance? I am also not entirely sure how relevant/helpful the theoretical analysis is. One of the main contributions of the paper seems to be about enhanced diversity with the proposed strategy, and yet there isn't any diversity involved in the section. It would be much more interesting if the authors of the paper can theoretically demonstrate the diversity aspect of the proposed method? Lastly, it seems that the proposed method also introduces additional hyper-parameters like the deviation rate. How sensitive is the performance of the algorithm to the choice of hyper-parameters, and how easy is it to tune for it?

Overall I think this is a good submission and recommend accepting it.

[1] Chang, Haw-Shiuan, Erik Learned-Miller, and Andrew McCallum. "Active bias: Training more accurate neural networks by emphasizing high variance samples." Advances in Neural Information Processing Systems 30 (2017): 1002-1012.

[2] Ren, Mengye, et al. "Learning to reweight examples for robust deep learning." International Conference on Machine Learning. PMLR, 2018.

**Time Spent Reviewing:**

2 hours

---

> ### Author Response · Authors · 2021-08-08
> **Response to Reviewer Bmco**
>
> We appreciate your feedback and suggestions to this paper. There is indeed a connection between our method and boosting, as the reweighting scheme is inspired by it, and that is why we name our method Checkpoint-Boosted Neural Networks. Also, your recommendations are quite interesting and inspiring literatures that might trigger further developments to our work. The followings are detailed responses to your concerns/suggestions:
>
> 1. *"As an illustrative purpose, It might be worthwhile to see how big the gap is in terms of accuracy and diversity when compared to an explicit ensemble."*
> &emsp;&emsp; We appreciate your suggestions very much. The explicit ensembles are expected to achieve higher accuracy with more expensive computational cost, and demonstrating this gap must be helpful to illustration. We will conduct experiments on this problem and attach the results to the final version of this paper if time allows. However, it may be very time-consuming to train the explicit ensembles, especially on the large-scale datasets such as ImageNet. On the other hand, such explicit ensembles often enhance the diversity by adopting different random initialization or training models on different sub datasets (for example, Bagging), which differ from the scheme of CBNN and may not lead to optimal diversity.
>
> 2. *"It might be interesting to also give a more careful analysis (in addition to the ones present in section 5.3) on why the proposed method outperforms previously proposed checkpoint ensemble methods. Is it because of larger diversity or better individual model performance? I am also not entirely sure how relevant/helpful the theoretical analysis is. One of the main contributions of the paper seems to be about enhanced diversity with the proposed strategy, and yet there isn't any diversity involved in the section. It would be much more interesting if the authors of the paper can theoretically demonstrate the diversity aspect of the proposed method?"*
> &emsp;&emsp; The diversity of base models (checkpoints) is very important for the predictive performance of ensembles. Actually, beside of measuring diversity by calculating pairwise correlations (see Section 5.3), there is also another way to analyze the diversity. Specifically, the difference of the model parameters, which can be measured by the Euclidean distance between the model parameter vectors of two checkpoints, also reflects the diversity. Intuitively, the models that are further from each other in the parameter space are expected to be more diverse. We visualize the loss surface in Fig.1 in the way introduced in Section 4.2, where the checkpoints we use (point p1, p2, and p3) lie in different local optima on the loss surface and keep distance with each other.
> &emsp;&emsp; Moreover, it is a good concern that if CBNN improves performance by enhancing diversity or improving individual models. Actually, we have omitted the test accuracy of the individual checkpoints in Table 1 to prevent redundancy. However, it might be helpful to figure out that CBNN improves the diversity, as the best-performing individual checkpoints in CBNN often yield slightly lower accuracy than  straightforwardly training single model does. The results (test error with standard deviation) are as following:
> | Model | Method          | CIFAR-10 | CIFAR-100         | T-ImageNet       | ImageNet         |
> | ----- | --------------- | -------- | ----------------- | ---------------- | ---------------- |
> | R-110 | Single Model    | 5.63     | 27.67 $\pm$ 0.25  | 45.10 $\pm$ 0.36 | -                |
> | R-110 | Best Checkpoint | 6.14     | 28.52 $\pm$ 0.26  | 45.59 $\pm$ 0.39 | -                |
> | D-40  | Single Model    | 5.13     | 23.15 $\pm$ 0.22  | 38.24 $\pm$ 0.34 | -                |
> | D-40  | Best Checkpoint | 5.58     | 23.55 $\pm$ 0.24  | 39.01 $\pm$ 0.37 | -                |
> | E-B0  | Single Model    | 3.88     | 20.87 $\pm$ 0.25  | 34.50 $\pm$ 0.35 | 23.70 $\pm$ 0.21 |
> | E-B0  | Best Checkpoint | 3.94     | 21.85 $\pm$ 0.26  | 34.88 $\pm$ 0.36 | 24.92 $\pm$ 0.25 |
> | E-B3  | Single Model    | 3.45     | 17.28 $\pm$ 0.22  | 31.86 $\pm$ 0.30 | 18.30 $\pm$ 0.15 |
> | E-B3  | Best Checkpoint | 3.69     | 18.65 $\pm$0 0.23 | 31.57 $\pm$ 0.32 | 19.42 $\pm$ 0.20 |
> 3. *"It seems that the proposed method also introduces additional hyper-parameters like the deviation rate. How sensitive is the performance of the algorithm to the choice of hyper-parameters, and how easy is it to tune for it?"*
> &emsp;&emsp; Right, deviation rate is a hyper-parameter we introduce in our proposed method, which represents the tradeoff between the diversity and accuracy of individual models. Very low deviation rates leads to low diversity of checkpoints, as setting deviation rate to zero is equivalent to omit the data reweighting process. Very high deviation rates (1 for example), however, are detrimental to convergence.
> &emsp;&emsp; It is robust to choose relatively low deviation rates, such as $0.01$, the value we recommended in our paper. However, since the sample reweighting process in our method is adaptive, there is very low impact on the performance if we set a slightly higher or lower deviation rate. For example, with the same settings in Table 1, the accuracy of CBNN with EfficientNet-B0 on CIFAR-100 only decrease by 0.35% and 0.28%, respectively, if setting $\eta=0.02$ and $\eta=0.005$ (higher than $0.02$ deviation rate is not recommended since Eq.8 may be violated or the number of checkpoints is limited).

---

> > ### Comment · Reviewer_Bmco · 2021-08-26
> > **Thank you for such detailed response!**
> >
> > Your response answered all of my questions. I particularly found the additional results provided above to be pretty insightful. Maybe you can add it in the Appendix of the paper.

---

> > > ### Author Response · Authors · 2021-08-26
> > > **Thank you for your reply!**
> > >
> > > We feel very happy to address your concerns about this paper! We will add the additional results in camera-ready version.

---

### Official Review · Reviewer_v3ks · 2021-07-19

**Rating:** 6
**Confidence:** 4

**Summary:**

The authors made a study on how to ensemble checkpoints with a boosting scheme, leading to faster convergence and more diversity. By adaptively reweight the training samples, Checkpoint-Boosted Neural Network (CBNN) can accelerate convergence and encourage diversity among individual ensemble members. The empirical evaluation demonstrated that under the same training budget, CBNN outperforms a number of baselines including Snapshot Ensemble. The authors also showed that it achieves great performance on imbalanced dataset because of CBNN's adaptability.

**Limitations And Societal Impact:**

The authors discussed the limitations.

**Main Review:**

Snapshot Ensemble requires a modification to the learning rate scheme. This limits its application when cylinder learning rate scheme is not desirable. CBNN does not require any strange tweak to the current learning algorithm, so it has a much wider application. The authors provide convergence analysis, leading to a theoretically grounded algorithm. The baselines this paper compared to are competitive, including recent advances such as Snapshot Ensemble and FGE. CBNN outperforms these competitive baselines on a variety of network architectures and vision datasets in terms of the accuracy measure. Additionally, the diversity analysis section is also interesting as it provides more insights of the CBNN. It also demonsrates that the proposed reweighting scheme is helpful. Overall, the empirical evaluation is convincing and is consistent to the intuition of the proposed method.

I also have a number of concerns about this paper. One of the major concern is that the paper is missing some ablation studies which are helpful to understand the proposed algorithm. For example, under the fixed training budget, how does the accuracy and diversity change if we change the training iterations per checkpoint t / or M. Another ablation study is dropping the sample reweighting during training but still keeping ensemble reweighting based on the error rate, how does this change the accuracy and diversity? This demonsrates the importance of sample reweighting which is the core component in the CBNN.

There are some other minor concers. For example, it might be better to report the accuracy and diversity (Figure 2a) of fully independent ensembles, which can be used as an upper bound. The gap also shows how much room left to improve CBNN. It is also benificial to report individual accuracy of each ensemble member (each checkpoint). This helps understand the diversity, too.

One of the best advantages of this algorithm is its wide applicability. The paper would be more convining if the authors can demonsrate its effectiveness in the trainsformer models in the NLP domain. Snapshot Ensemble and FGE are naturally short in these applications. Another potential experiments are uncertainty estimation, deep ensembles are powerful not only because of its improved accuracy but only the improved calibration / uncertainty quantification. The paper can be stronger if the authors can show CBNN has better calibration (expected calibration error,  ECE) performance than the single model.

Overall, I consider this paper is slightly above boarderline at its current shape.

**Time Spent Reviewing:**

4

---

> ### Author Response · Authors · 2021-08-08
> **Response to Reviewer v3ks**
>
> We appreciate your careful review and feedback on this paper. As what you have mentioned, the superior predictive performance, higher efficiency, as well as the wider application are main strengths of CBNN, compared with the existing checkpoint ensemble methods. We are also exploring its broader use in NLP, in particular, the transformer models, as future works.
>
> The following responses might be helpful to address your concerns about this paper:
>
> 1. *"The paper is missing some ablation studies which are helpful to understand the proposed algorithm. For example, under the fixed training budget, how does the accuracy and diversity change if we change the training iterations per checkpoint t / or M. Another ablation study is dropping the sample reweighting during training but still keeping ensemble reweighting based on the error rate, how does this change the accuracy and diversity? "*
> &emsp;&emsp; We appreciate your advice and reminder very much. It is a very good suggestion to take sufficient ablation studies, including the two you mentioned. Actually, these two mentioned ablation studies have already been included in our technical appendix (in Supplementary Material), or seen as special cases in the experimental results.
> &emsp;&emsp; The former study, that what if we change the number of checkpoints under the fixed training budget, has actually been done and presented in the technical appendix (see Fig.3 in Appendix). In that part, we manually change the number of checkpoints, i.e., $M$, with a fixed training budget and then compare the accuracy of CBNN with the baselines. The results, shown in Fig.3, demonstrate the superior performance of CBNN over the baselines. Also, it indicates that CBNN continues to yield better performance as $M$ increases, while the baselines, such as SSE, may suffer from an accuracy decrease.
> &emsp;&emsp; The later study, dropping the sample reweighting during training but still keeping ensemble reweighting based on the error rate, is actually a special case of CBNN. This is equivalent to setting deviation rate $\eta=0$ (i.e., according to Eq.6 and Eq.7, the sample weights will not change). Therefore, although we did not directly present the results with $\eta=0$, we have mentioned that the presented experiment results in our paper (in Table 1, 2, and 3) are based on optimal or nearly optimal hyper-parameters. In other words, setting $\eta=0.01$ yields better results than setting it to zero. We will add additional notations about this question in the camera-ready version.
>
> 2. *"It might be better to report the accuracy and diversity (Figure 2a) of fully independent ensembles, which can be used as an upper bound. The gap also shows how much room left to improve CBNN. It is also beneficial to report individual accuracy of each ensemble member (each checkpoint)."*
> &emsp;&emsp; We appreciate your suggestions very much. The fully independent ensembles are expected to achieve higher predictive performance with more expensive computational cost, and reporting this gap must be helpful to better illustration. We will conduct experiments on this problem and attach the results to the final version of this paper if time allows. However, it may be very time-consuming to train the fully independent ensembles, especially on the large-scale datasets such as ImageNet. On the other hand, such fully independent ensembles often enhance the diversity by adopting different random initialization or training models on different sub-datasets (for example, Bagging), which differ from the scheme of CBNN and may not lead to optimal diversity.
> &emsp;&emsp; We also pay attention to individual networks of CBNN. Actually, we have omitted the test accuracy of the individual checkpoints of CBNN in Table 1 to prevent redundancy. However, it might be helpful to figure out  if CBNN improves performance by enhancing diversity or improving individual models. As presented in the following table (the values in the table denote test error rate $\pm$ standard deviation), the best-performing individual checkpoints in CBNN often yield slightly higher test error than straightforwardly training single model does. We will add these experiment results into Table 1 in camera-ready version.
> | Model | Method          | CIFAR-10 | CIFAR-100        | T-ImageNet       | ImageNet         |
> | ----- | --------------- | -------- | ---------------- | ---------------- | ---------------- |
> | R-110 | Single Model    | 5.63     | 27.67 $\pm$ 0.25 | 45.10 $\pm$ 0.36 | -                |
> | R-110 | Best Checkpoint | 6.14     | 28.52 $\pm$ 0.26 | 45.59 $\pm$ 0.39 | -                |
> | D-40  | Single Model    | 5.13     | 23.15 $\pm$ 0.22 | 38.24 $\pm$ 0.34 | -                |
> | D-40  | Best Checkpoint | 5.58     | 23.55 $\pm$ 0.24 | 39.01 $\pm$ 0.37 | -                |
> | E-B0  | Single Model    | 3.88     | 20.87 $\pm$ 0.25 | 34.50 $\pm$ 0.35 | 23.70 $\pm$ 0.21 |
> | E-B0  | Best Checkpoint | 3.94     | 21.85 $\pm$ 0.26 | 34.88 $\pm$ 0.36 | 24.92 $\pm$ 0.25 |
> | E-B3  | Single Model    | 3.45     | 17.28 $\pm$ 0.22 | 31.86 $\pm$ 0.30 | 18.30 $\pm$ 0.15 |
> | E-B3  | Best Checkpoint | 3.69     | 18.65 $\pm$ 0.23 | 31.57 $\pm$ 0.32 | 19.42 $\pm$ 0.20 |

---

> > ### Comment · Reviewer_v3ks · 2021-08-30
> > **Re: Discussion**
> >
> > Thanks for submitting the author feedback. The additional experiments addressed my concerns about the ablation study. My concerns regarding experiments scope (the wide applicability claim) still remain. So I keep my score as weak acceptance.

---

> > > ### Author Response · Authors · 2021-08-31
> > > **Thank you for your reply!**
> > >
> > > It is glad to address your concerns. We are also exploring the wider application of our algorithm. Again, thank you very much for your detailed comments!

---

### Official Review · Reviewer_Ca6w · 2021-07-23

**Rating:** 6
**Confidence:** 4

**Summary:**

This paper proposed to ensemble the checkpoint models during the training to boost the performance of neural networks. To improve diversity over checkpoints, the proposed method changes the weights for each sample based on the mis-classification rate along the training process. When finally doing the ensemble over the checkpoints, a weight for each checkpoint based on mis-classification rate is also assigned. The experimental results show that comparing with existing checkpoint-based ensemble models and single model,  the proposed method has 1) better accuracy; 2) high diversity over different checkpoints; 3) low training cost.

**Limitations And Societal Impact:**

I don't think the paper has explicitly mentioned limits of the methods.
The paper showed better performance on imbalanced datasets in Sec. 5.4. Since imbalanceness is common in datasets with minority groups, it would be good if the authors can discuss the impact of their method in this aspect.


**Main Review:**

Overall, the paper is well-written and easy to follow. The proposed method is straightforward and should be novel. The experimental results look promising on both performance and training cost.
But I have a few questions and concerns.

1. Is there a reason why ImageNet with Densenet and Resnet are omitted in Tab. 1?
2. As SSE is the second-best performance, I am wondering if the authors tried to adopts cyclic learning rate schedule as SSE does to improve CBNN?
3. The theoretical analysis Theorem 1 and Theorem 2 seem trivial and the conclusion is a little bit vacuous. In particular, the conclusion is that a upper bound of the exponential loss is decreasing with more checkpoints. The upper bound can be very loose, which can lead to a vacuous conclusion. It would be better to calculate the tightness of the upper bound empirically.
4. Although random labeling has the error rate (k-1)/k, the worst case error rate can be worse than this. Therefore, $\lambda_m > 0$ needs to be assumed.

Overall, as the experimental results look promising, I think this paper would be interesting to the DL community. However, due the above concerns and lack of the limitation and societal impact discussion, I think the current version of the paper is not ready for publication. I am open to change the score based on the authors' rebuttal.

**Time Spent Reviewing:**

2

---

> ### Author Response · Authors · 2021-08-08
> **Response to Reviewer Ca6w**
>
> We appreciate your careful comments and constructive suggestions.  The followings are detailed responses to your questions/concerns:
>
> 1. *"Is there a reason why ImageNet with Densenet and Resnet are omitted in Tab. 1?"*
> &emsp;&emsp; As originally reported in [1] and [2], both ResNet and DenseNet have two different sets of standard architectures for CIFAR and ImageNet, respectively. The architectures for CIFAR (e.g., ResNet-110 and DenseNet-40) are quite different from those for ImageNet (e.g., ResNet-50, ResNet-101, DenseNet-121, and DenseNet-169). Specifically, the architectures for CIFAR have relatively small number of parameters and require 32x32 input size, while the architectures for ImageNet require that size of 224x224. This difference has been discussed in detail in Section 4 of [1] and Section 3 of [2], respectively.
> &emsp;&emsp; As the 110-layer ResNet and the 40-layer DenseNet used in our paper are originally designed for CIFAR, they are rather suitable to the low-resolution datasets such as CIFAR-10, CIFAR-100, and Tiny-ImageNet. These two networks are not expected to achieve high accuracy on ImageNet, due to their small model size and input size. In addition, to evaluate our method on ImageNet, we choose to use EfficientNet-B0 and EfficientNet-B3 [3], as these networks are able to achieve higher accuracy and more efficient in computation and memory usage, compared with ResNet-50 and DenseNet-121.
> [1] Kaiming He, Xiangyu Zhang, Shaoqing Ren, and Jian Sun. Deep residual learning for image recognition. In Proceedings of the IEEE conference on computer vision and pattern recognition, pages 770–778, 2016.
> [2] Gao Huang, Zhuang Liu, Laurens Van Der Maaten, and Kilian Q Weinberger. Densely connected convolutional networks. In Proceedings of the IEEE conference on computer vision and pattern recognition, pages 4700–4708, 2017b.
> [3] Mingxing Tan and Quoc Le. Efficientnet: Rethinking model scaling for convolutional neural networks. In International Conference on Machine Learning, pages 6105–6114, 2019.
>
> 2. *"As SSE is the second-best performance, I am wondering if the authors tried to adopts cyclic learning rate schedule as SSE does to improve CBNN?"*
> &emsp;&emsp; We thank you for your suggestion. Actually, we have already tried several experiments by combining SSE with CBNN. Unfortunately, they incurred poor performance, and sometimes even yielded worse accuracy than Single Model (Training the network without ensemble). This might be because the cyclic learning rate in SSE is not well applicable to CBNN. As there is a rapid increase of learning rate between the cycles, which meanwhile leads to high training error. This high error will mislead the update of sample weights in CBNN and thereby impact the performance. Specifically, with the high training error, there will be many misclassified samples (i.e., $G_m^{y_i}(x_i)=0$) in Eq.6. Therefore, according to Eq.6 and Eq.7, the weights of these misclassified samples will increase and the weights of the correctly classified samples will decrease rapidly due to the normalization ($Z_m$ in Eq.6 and Eq.7 is a normalization term to keep $\sum_{i=1}^n{\omega_{mi}}=1$).
>
> 3. *"The theoretical analysis Theorem 1 and Theorem 2 seem trivial and the conclusion is a little bit vacuous. In particular, the conclusion is that a upper bound of the exponential loss is decreasing with more checkpoints. The upper bound can be very loose, which can lead to a vacuous conclusion. It would be better to calculate the tightness of the upper bound empirically."*
> &emsp;&emsp; We appreciate your comments on the theoretical analyses very much. The analyses in Section 4.1, including Theorem 1 and Theorem 2, explain how CBNN benefits model's performance from the perspective of ensemble. These analyses serve not only to demonstrate the loss decrease, but also as an explanation of our motivations and the settings of some hyper-parameters such as deviation rate.
> &emsp;&emsp; Formally, according to Theorem 1 and Theorem 2 (Formula 11 and Formula 12), we have  $L_{exp} \leq \prod_{m=1}^M{Z_m}$, $Z_m<1$. Therefore, as we introduce more checkpoints (i.e., $M$ increases), the upper bound of $L_{exp}$ decreases exponentially, where the decreasing rate depends on how small the $Z_m$ is. Note that, it can be inferred from Formula 12 that $Z_m$ is smaller with lower training error $e_m$. Taking $\lambda=0.01$ on CIFAR-100 as an example, $Z_m=0.9409$ when $e_m=0.1$, which indicates that the upper bound of $L_{exp}$ will drop by $0.9409$ if we introduce such a checkpoint (so, if we introduce $10$ checkpoints with $e_m=0.1$, this bound will drop by $0.9409^{10}=0.5438$).
> &emsp;&emsp; Therefore, Section 4.1 is relatively a loose analysis as there is no additional model assumptions. For example, if we assume that the training error is upper bounded as 20% (actually most networks on CIFAR-10 and CIFAR-100 in our experiments satisfy this assumption), the bound of $L_{exp}$ can be much tighter.
> &emsp;&emsp; Specifically, assume that the error rate of each model satisfies $e_m<e_{max}$, so we have $\lambda_m > \lambda_{min}=\log((1-e_{max})/e_{max})+log(k-1)$. Thus, according to Theorem 1 and Theorem 2 (Formula 11 and Formula 12), we have $L_{exp} \leq \prod_{m=1}^M{Z_m} \leq \prod_{m=1}^M{[(1-e_m)\mbox{exp}(-\eta\lambda_m)+e_m]} \leq [(1-e_{max})\mbox{exp}(-\eta\lambda_{min})+e_{max}]^M$. Therefore, taking $M=10$, $\eta=0.01$ on CIFAR-100 as an example, the upper bound of $L_{exp}$ will be $0.6215$ if $e_{max}=0.2$, while this bound is $0.9766$ if $e_{max}=0.9$.
>
> 4. *"Although random labeling has the error rate $(k-1)/k$, the worst case error rate can be worse than this. Therefore, $\lambda_m>0$ needs to be assumed."*
> &emsp;&emsp; Theoretically, we agree that the error rate in a $k$-class problem can be worse than $(k-1)/k$. However, in practical implementations, it is unnecessary to theoretically assume that $e_m<(k-1)/k$ or $\lambda_m>0$, since we can guarantee such conditions by the tricks that automatically reverse the model's predictions.
> &emsp;&emsp; Specifically, we first take a binary classification task as example and explain how this trick works. In our proposal, we save a checkpoint as base model before evaluating its training error rate. Therefore, in a binary classification task, when the evaluation result of error rate on the training set is higher than $0.5$ ($0.6$ for example), we can simply reverse the model's 0-1 prediction and save the reversed model as checkpoint, which has an error rate of $0.4$. Note that we can reverse the model's 0-1 prediction simply by attaching an additional layer of $y_r=1-y_o$ to the model, where $y_o$, $y_r$ denote the original and reversed prediction, respectively. Therefore, by adopting such automatic tricks, the error rate $e_m>(k-1)/k$ in binary classification tasks is equivalent to $1-e_m$.
> &emsp;&emsp; Similarly, in multi-class tasks ($k>2$), we can also utilize such tricks when $e_m>(k-1)/k$. The difference is that, in such cases, we need to evaluate the model at most $k-1$ times. For example, in a $k$-class task where labels lie in {$0, 1,\dots, k-1$} and we denote the model's original prediction as $y_o$. When getting a model with the error rate $e_m>(k-1)/k$, we can reverse the model by implementing $y_r=(y_o+n)\mbox{ Mod }k, \mbox{ for } n=1,2,\dots,k-1$ until $y_r$ yields the error rate $e_m<(k-1)/k$, where $y_r$ denotes the reversed prediction. It can be proved that there must be $e_m<(k-1)/k$ within the $k-1$ trials.
> &emsp;&emsp; Therefore, we can add such automatic mechanisms to guarantee the lower than $(k-1)/k$ error and don't have to assume $\lambda_m>0$ in theory, though such mechanisms are unnecessary in most practical cases, since the models we trained often yield much lower error than $(k-1)/k$. We appreciate your comments on this question very much and will add discussions about the tricks above in camera-ready version.
>
> 5. *"Imbalanceness is common in datasets with minority groups, it would be good if the authors can discuss the impact of their method in this aspect."*
> &emsp;&emsp; We appreciate your advice to link CBNN's capability of tackling imbalanceness to its potential social impact of minority groups. We will add this discussion in camera-ready version.

---

> > ### Comment · Reviewer_Ca6w · 2021-08-20
> > **Most of my concerns are addressed. Raise my score.**
> >
> > I'd like to raise my score based on the authors' response.

---

> > > ### Author Response · Authors · 2021-08-21
> > > **We are happy that our responses help to address your concerns**
> > >
> > > We are happy that the responses help to address your concerns. Again, we thank you very much for your positive comments.

---

> > > ### Author Response · Authors · 2021-08-30
> > > **We appreciate your decision to raise the score. However, it seems unchanged yet.**
> > >
> > > We thank you very much for your detailed comments and your decision to raise the score. However, it seems that the score remains unchanged. Would you mind to check it, since the discussion period is going to close at the start of September. We are sorry to disturb again.

---

> > > > ### Comment · Reviewer_Ca6w · 2021-08-30
> > > > **Score changed.**
> > > >
> > > > Sorry about that. Just changed.

---

> > > > > ### Author Response · Authors · 2021-08-30
> > > > > **Thanks a lot!**
> > > > >
> > > > > Thank you for your time!

---

### Decision · Program_Chairs · 2021-09-27

**Decision:**

Accept (Poster)

**Comment:**

I recommend to accept this paper.

In this paper, the authors proposed a boosting method to ensemble checkpoints during the training of neural networks called Checkpoint-Boosted Neural Network (CBNN) to improve the performance. In particular, a boosting scheme with both theoretical guarantee and empirical justification is proposed to accelerate model convergence and maximize the checkpoint diversity. In the post-rebuttal discussion, all the reviewers are agree that most concerns in the original reviewers are properly addressed. Reviewer Ca6wKai also raises the score. I will suggest the authors to take the suggestions from reviewers into account in the preparation of camera ready.